# TP53 Alterations in Myelodysplastic Syndromes and Acute Myeloid Leukemia

**DOI:** 10.3390/biomedicines11041152

**Published:** 2023-04-11

**Authors:** Ramy Rahmé, Thorsten Braun, James J. Manfredi, Pierre Fenaux

**Affiliations:** 1Department of Oncological Sciences and Tisch Cancer Institute, Icahn School of Medicine at Mount Sinai, New York, NY 10029, USA; ramy.rahme@aphp.fr (R.R.); james.manfredi@mssm.edu (J.J.M.); 2Institut de Recherche Saint Louis (IRSL), INSERM U1131, Université Paris Cité, 75010 Paris, France; 3Ecole Doctorale Hématologie–Oncogenèse–Biothérapies, Université Paris Cité, 75010 Paris, France; 4Clinical Hematology Department, Avicenne Hospital, Assistance Publique-Hôpitaux de Paris (AP-HP), Université Sorbonne Paris Nord, 93000 Bobigny, France; thorsten.braun@aphp.fr; 5The Graduate School of Biomedical Sciences, Icahn School of Medicine at Mount Sinai, New York, NY 10029, USA; 6Senior Hematology Department, Saint Louis Hospital, Assistance Publique-Hôpitaux de Paris (AP-HP), Université Paris Cité, 75010 Paris, France

**Keywords:** mutant p53, cancer, myelodysplasia, acute myeloid leukemia

## Abstract

TP53 mutations are less frequent in myelodysplastic syndromes (MDS) and acute myeloid leukemia (AML) than in solid tumors, except in secondary and therapy-related MDS/AMLs, and in cases with complex monosomal karyotype. As in solid tumors, missense mutations predominate, with the same hotspot mutated codons (particularly codons 175, 248, 273). As TP53-mutated MDS/AMLs are generally associated with complex chromosomal abnormalities, it is not always clear when TP53 mutations occur in the pathophysiological process. It is also uncertain in these MDS/AML cases, which often have inactivation of both TP53 alleles, if the missense mutation is only deleterious through the absence of a functional p53 protein, or through a potential dominant-negative effect, or finally a gain-of-function effect of mutant p53, as demonstrated in some solid tumors. Understanding when TP53 mutations occur in the disease course and how they are deleterious would help to design new treatments for those patients who generally show poor response to all therapeutic approaches.

## 1. Introduction

The transcription factor p53, encoded by the TP53 gene in humans, is one of the most studied tumor suppressor proteins. In normal cells, p53 activity is low, but in response to DNA damage and numerous other stress signals, p53 levels rise dramatically and result in the activation and transcription of genes with important roles in cell cycle arrest, senescence, apoptosis, metabolism, and differentiation [1]. The sum of these activities is to ensure that an abnormal cell fails to arise and proliferate. During normal hematopoiesis, p53 activities preserve genome integrity and regulate several cellular processes that maintain the normal stem cell pool and serve as a barrier to tumorigenesis [2,3].

Perturbations in p53 activity or p53-dependent pathways are required for the development of most cancers [4], and there is evidence in many situations to suggest that the restoration or reactivation of physiological p53 function may be therapeutic [5,6,7,8]. Various mechanisms are responsible for the disruption of p53 activity in cancer, mainly deletion or mutation of the TP53 gene, and overexpression of the p53 negative regulators Mdm2 and Mdm4 [9,10,11]. Several isoforms are encoded by the TP53 gene and appear to play different roles in tumorigenesis/cancer progression [12] or response to treatment, for example in acute myeloid leukemia (AML) [13]. Regardless of the mechanism behind p53 dysfunction, the downstream consequences are profound due to the very large spectrum of biological activities in which p53 is normally implicated. The clinical correlation between p53 mutational status in cancer cells and resistance to treatment has been studied since the 1990s. In breast cancer, presence of a TP53 mutation is associated with resistance to doxorubicin [14,15]. Similarly, ovarian cancer patients harboring a TP53 mutation are less sensitive to treatment with cisplatin [16,17]. Moreover, the association of TP53 mutation with chemoresistance and poor prognosis has also been observed in lung [18], gastric, and colorectal cancers [19], as well as in hematological malignancies [20,21]. Apart from triggering chemoresistance, p53 mutants are also able to attenuate cancer response to radiotherapy [14,19,22].

Myelodysplastic syndromes (MDS) are a clonal hematopoietic stem cell disorders characterized by cell dysplasia, ineffective hematopoiesis that leads to cytopenias (mainly anemia) and a variable risk of progression to AML [23]. Furthermore, AML is characterized by clonal expansion of undifferentiated myeloid precursors, resulting in impaired hematopoiesis and life-threatening cytopenias. Among the prognostic factors identified in both malignancies, presence of a mutation in the TP53 gene indicates a particularly dismal prognosis irrespective of the treatment administered [24]. In this review, we present an overview of the current knowledge about mutant p53 activities in MDS/AML, and discuss treatment strategies that may potentially enter clinical testing.

## 2. Characteristics of TP53 Mutations in MDS/AML

TP53 gene mutations are reported in about 50% of solid tumors, but only 5–10% of de novo myelodysplastic syndromes (MDS) and acute myeloid leukemia (AML). However, while these mutations are rare in MDS/AML with a normal karyotype (1%), they are seen in 40–50% of secondary and therapy-related cases [20,25,26,27], and in 80% of complex monosomal karyotypes (CK) that include 17p and/or 5q deletion [28,29]. Other inactivating mechanisms of wild-type p53 function include overexpression of MDM2 and MDM4, negative regulators of p53, in 20–30% and 40–50% of AML cases, respectively [30,31,32,33,34]. Unfortunately, treatment with Mdm2 inhibitors has not so far demonstrated a substantial effect in these cases [35]. Inactivation of p14-ARF, a positive regulator of p53, has been more rarely reported in AML [32,36,37]. 

As in solid tumors, most TP53 mutations in MDS/AML cluster in exons 4 to 8 that encode the DNA binding domain [38,39]. Nearly 75% of the mutants are missense: recurrent hotspot variants are also observed, as in solid tumors, and at the same frequent codons such as the contact mutants at codons 248 and 273, and the structural mutants at codons 175 and 245 [40]. The number of cooperating driver mutations is usually very low in TP53-mutated MDS/AML, and even absent in 85% of the cases [41]. 

TP53 alteration in MDS/AML can be monoallelic or biallelic [42]. Monoallelic mutations, with point mutation of only one TP53 allele, represent 25 to 30% of the cases. They are particularly seen in MDS with isolated 5q deletion [43,44,45,46], but can also be discovered in various MDS and AML with non-complex karyotype by systematic NGS analysis, generally at low variant allelic frequency (VAF) [42]. In 70 to 75% of the cases, TP53 alterations are biallelic, generally resulting from point mutation on one TP53 allele (usually missense) and loss of the other allele, through 17p deletion or monosomy 17. The great majority of MDS/AMLs with biallelic alteration have indeed complex monosomal karyotype resulting in 17p deletion, very often with 5q deletion [47]. On the other hand, monoallelic TP53 mutations can lead to genomic instability, chromosomal loss and chromothripsis (also known as “chromosome shattering” that leads to large structural rearrangements in chromosomes [48]) including loss of heterozygosity (LOH) through 17p deletion [49], and therefore resulting in a biallelic hit. TP53 mutations, especially in the biallelic state, are associated with resistance to most treatments, including chemotherapy, hypomethylating agents, and even allogeneic stem cell transplantation, as the risk of relapse post-transplant is very high, at least in case of biallelic alterations [50,51,52,53]. Monoallelic mutations are also, to a lesser extent, associated with a certain resistance to treatment, for example to lenalidomide–an immuno-modulatory drug used in MDS with del(5q) [43,54], but have generally a more limited impact on survival. 

## 3. When Do TP53 Mutations Arise in MDS and AML Cells?

Whether TP53 mutations occur early or late in the evolution of MDS and AML remains uncertain in most situations. In well-performed studies in therapy-related MDS/AML, it was found that cells harboring the leukemia-specific TP53 mutation preexisted to the first cancer and were positively selected by treatment for this cancer (chemotherapy or radiation) to which they were resistant [55,56]. Progressive evolution was seen through genomic instability and chromothripsis with inactivation of the second TP53 allele, as seen above [49] (Figure 1). TP53 mutations are also among the mutations found in healthy individuals with clonal hematopoiesis of indeterminate potential (CHIP), who have an increased risk of developing various blood cancers [57,58]. However, DNMT3A, TET2, and ASXL1 mutations largely predominate in CHIPs seen in patients with no prior history of treated cancer, and CHIPs with TP53 mutation are mainly seen in patients having received chemotherapy or radiotherapy for a prior cancer, where they increase the risk of therapy-related MDS/AML, potentially by the selection mechanism just described above [55,59,60,61,62,63,64]. 

It is still unclear in other MDS/AML situations when TP53 mutations arise. In MDS with isolated 5q deletion, detectable monoallelic TP53 mutations, often at very low VAF and sometimes multiple, are found in 20% of the patients at diagnosis [43,44,45,46]. Whether they occur during disease evolution or very early in minor undetectable clones (by conventional methods) that could undergo positive selection, including by treatment with lenalidomide, is unclear. In those MDS with isolated del(5q) and monoallelic TP53 mutation, as in the therapy-related model described above, progression to AML is generally preceded by acquisition of a complex karyotype, including del(17p) and biallelic TP53 inactivation [65,66] (Figure 2). 

In myeloproliferative neoplasms, TP53 mutations are seen in about 20% of the cases with progression to MDS/AML. They appear to be late events, although once again small TP53 mutant clones, undetectable by conventional techniques, may have occurred earlier in the disease course [67,68,69,70,71,72,73]. When TP53-mutated clones appear during the disease course would be important to determine, as their early detection in myeloid malignancies could allow to target them pharmacologically, potentially preventing progression to full-blown biallelic TP53 MDS/AML. 

TP53 mutations could represent early leukemogenic events in many situations, and they have been indeed reported in pre-leukemic hematopoietic stem cells (HSCs) of AML patients [74,75]. Major roles of TP53 are related to cell-cycle control, DNA repair and apoptosis. Dysregulation of these pivotal functions might be an alternative mechanism to epigenetic modifications in establishing a proleukemogenic state in HSCs [76]. By transforming HSCs into pre-leukemic stem cells (pre-LSCs), TP53 mutations substantially contribute to the development of AML and its resistance to conventional treatments [55,77,78]. Moreover, TP53-mutated pre-LSCs retain their ability to differentiate into mature blood cells both in patient-derived mouse xenografts and patients with AML [78]. Clonogenic assays revealed that patient-specific TP53 mutations are present in the vast majority of HSC-derived colonies (median, 97%; range, 45 to 100%), with only a paucity of cooperating mutations in known cancer genes, as seen above. Copy-number alterations, as already described, appear to be secondary events after the onset of a first TP53 mutation [79,80]. 

## 4. How Can p53 Mutants Drive Leukemia Development and Maintain the Leukemic State?

Mutant p53 proteins can potentially contribute to MDS/AML pathophysiology and progression through different molecular mechanisms: protein inactivation and loss of function in the case of biallelic hits; a dominant-negative effect of the mutant over germline wild-type p53; or finally a gain-of-function effect of the mutant protein. Those different mechanisms are not exclusive. In the last two situations, monoallelic mutations may already be pathogenic. The fact that poor prognosis in MDS/AMLs with TP53 mutation is mainly seen in patients with biallelic alterations [42]—where MDS/AML cells have no more functional p53–intuitively suggests that loss of wild-type p53 functions is the major oncogenic determinant, at least in late disease stages.

In tumors in general, the predominance of mutant p53 protein expression over the simple loss of p53, given that missense mutants overshadow nonsense/truncating mutants, suggests an inherent biological advantage for TP53 mutants in human cancers [81,82,83]. This is in contrast to many other tumor suppressor genes that undergo deletion through the course of tumor initiation or development, such as PTEN, BRCA1, and RB1 [84]. Thus, because p53 normally acts as a tetramer, p53 mutants may play a dominant-negative effect on the activity of any remaining wild type p53 in the cancer cell [85,86]. In addition, missense mutations could also result in the gain of novel functions that contribute to tumorigenesis [87,88,89,90].

Whereas many scientific works have demonstrated mutant p53 activities in solid tumors [91,92,93], only few studies have addressed these functions in AML. Boettcher et al. showed in engineered human cell lines that gain-of-function (GOF) activities might be dispensable in AML [94], and that the dominant-negative expression (DNE) of TP53 mutants appeared to be the main driver of leukemogenesis, leading to p53 inactivation. However, in that work, TP53 mutations were introduced in AML lines that already depended on the activity of fusion oncoproteins, a situation where GOF activities of the mutants may not be essential for the cells. They also reported that missense mutations did not worsen prognosis in AML patients treated with intensive chemotherapy compared to patients with truncating mutations, adding another argument against any GOF in AML. However, intensive chemotherapy may not be the best treatment option in TP53-mutated patients, and these findings may therefore not be sufficient to rule out some GOF activities of missense mutants in AML. Therefore, AML murine models with various TP53 mutations are developed, including by our group, in order to avoid potential biases associated with cell line-based studies (work in progress). 

Furthermore, the Scott Lowe Lab developed an original RNA interference-based approach to generate murine AMLs by knocking down the expression of the two tumor suppressors Mll3 and Nf1, thus partly mimicking 7q and 17p deletions found in human AMLs, respectively (“MNP” model for Mll3, Nf1 and P53) [95]. This model was used to generate murine p53-mutated AML harboring the missense mutant R172H (equivalent to the human R175H hotspot mutant) [96]. In that setting, mutant p53 showed GOF activities through the pluripotency factor Foxh1 [96]. In line with those results, our in vivo studies also revealed GOF transcriptional activities for mutant p53 that conferred a proliferation advantage over p53 inactivation (manuscript in preparation). In a follow-up paper, Liu et al., using the MNP model on HSCs harboring the 11B3 chromosomal deletion–a 4 Mb region on mouse chromosome 11 that is syntenic to human 17p13.1 [97], showed that 11B3 deletion promoted the development of more aggressive AML compared to the p53-null background, suggesting that other genes than TP53 drive the selection for 17p loss during leukemogenesis [97]. 

## 5. Treatment of TP53 Mutated MDS/AML 

### 5.1. The Current Disappointing Situation 

All clinical studies have shown the poor outcome of TP53-mutated MDS/AML cases with current therapeutic approaches including intensive chemotherapy, hypomethylating agents (HMAs, i.e., azacitidine and decitabine), with or without other agents, and allogeneic stem cell transplantation, at least in the case of biallelic alterations that are generally associated with complex monosomal karyotypes–which constitutes the most frequent situation. 

Both missense and truncating TP53 mutations confer resistance to treatment in AML patients. TP53 mutation appears to be, even in complex karyotypes, an independent poor prognostic factor [50,51]. In addition, in low-risk MDS with 5q deletion [43], monoallelic TP53 mutations are associated with lower rates of response to lenalidomide and a higher risk of progression to AML (generally associated with acquisition of a complex karyotype, and loss of the second TP53 allele) [98]. Hence, the management of TP53-mutated MDS/AMLs is still highly challenging [51,52,53]. 

Intensive chemotherapy yields very low complete remission rates in AML with (generally) biallelic TP53 hits (around 30%), and prolonged myelosuppression [66,99]. HMAs [51,53] and more recently combinations of HMA and Venetoclax [100], a highly selective bioavailable inhibitor of the anti-apoptotic BCL2 protein, have increased the response rate in MDS/AML with complex monosomal karyotypes and biallelic TP53 hits (up to 50–60%), but those responses are short and median survival remains at around one year. Finally, allogeneic stem cell transplantation is associated with a very high risk of relapse post-transplant in these patients [101,102], but its results are better in patients with monoallelic mutations, especially patients with isolated 5q deletion [103]. 

### 5.2. How Can Treatment of MDS/AMLs with TP53 Mutation Be Improved? 

Many agents are being considered and tested to overcome the resistance of TP53-mutated cells to conventional treatments. A non-exhaustive list can be found in Table 1. They generally aim at inducing tumor cell death in the absence of a functional p53 protein or at reconforming mutant p53 protein to a functional state, although some may implicate the immune system such as CD47 blockers. Many of these agents have so far been only tested in vitro. In some cases, those drugs are used in other therapeutic indications–hence, they have already demonstrated a good toxicity profile–and could potentially be “repurposed” for the treatment of TP53-mutated MDS/AMLs. We will focus on a few of those drugs, stressing the fact that, so far, none of those agents has demonstrated a significant benefit in randomized phase III studies.

### 5.3. Drugs Currently Used in Clinical Trials

APR-246 (Eprenetapopt): A variety of molecules have been developed through which a native-like function of mutant p53 can be reactivated. Only one of those molecules, APR-246, has reached the clinical trial stage. APR-246 induces the refolding of mutant p53 in vitro and in mice, which enhances its DNA binding ability, resulting in the upregulation of wild-type p53 target genes and the induction of cell death by apoptosis [5,104]. Mechanistically, the decomposition products of APR-246, specifically the compound MQ (*methylene quinuclidinone*), covalently associate and form adducts with thiols of p53 mutants such as R175H, R248Q, and R273H [105]. Studies also uncovered a mutant p53-independent effect of APR-246: it depletes glutathione content and sensitizes mutant p53-containing cancer cells to oxidative stress [106,107]. This molecule was evaluated in ovarian carcinoma in combination with chemotherapy [108], and in combination with azacitidine for the treatment of TP53-mutated high-risk MDS and AML [109,110]. Two phase II trials of APR-246 plus azacitidine showed promising results [109,110], but the phase III trial against azacitidine monotherapy did not show a significant improvement in both complete remission and survival rates (*p* = 0.13). 

Magrolimab: CD47 is a cell surface protein belonging to the immunoglobulin family, expressed by virtually all cells in the body [111]. The interaction of CD47 with SIRPα on the surface of phagocytic cells gives a “do not eat me” signal to the innate immune system: this system is used by cancer cells to evade detection and destruction by macrophages [112]. CD47 was found to be highly expressed on AML cells [113] and to be associated with adverse prognosis [114]. Magrolimab is a monoclonal antibody that targets CD47. A first-line treatment with the triple combination of azacitidine, venetoclax, and magrolimab gave promising results in patients with TP53-mutated AML: overall response rate at 74% and complete remission rate at 41% [115]. A phase III randomized trial comparing azacitidine/venetoclax/magrolimab to azacitidine/venetoclax is currently enrolling (ENHANCE-3, NCT05079230). 

### 5.4. Drugs so Far Used Only in Preclinical Models

MCL1 inhibition: Myeloid Cell Leukemia 1 (MCL-1) protein is a member of the Bcl2 family of anti-apoptotic proteins (which also includes Bcl-2, Bcl-W and Bcl-xL). Through their BH3 domains, these proteins, in particular Mcl-1 and Bcl-xL, bind and sequester the pro-apoptotic proteins Bak and Bax on the mitochondrial outer membrane, thus preventing their activation [116]. In AML cell lines and patient-derived xenografts, Thijssen et al. found that TP53 deficiency altered the cell balance in BCL2 family members [117]. Upon exposure to the Bcl-2 inhibitor venetoclax, the absence of apoptosis after engagement of mitochondrial outer membrane permeabilization (MOMP) led to a state termed “Minority MOMP” that is deleterious and paradoxically promotes DNA damage and genomic instability [118,119]. This state further enhances the fitness of TP53-deficient cells, their competition ability and resistance to drugs. Combined treatment of venetoclax with an MCL1 inhibitor successfully induced apoptosis that improved the survival of mice engrafted with three different TP53-deficient xenografts [117]. 

Arsenic trioxide: A recent work showed that arsenic trioxide (ATO), a drug used in the treatment of acute promyelocytic leukemia, can rescue the conformation of multiple structural p53 mutants [120]. Mechanistically, ATO interacts with the four sulfur-containing p53 residues (Cys124, Cys135, Cys141, and Met133) that define the arsenic-binding pocket (ABP). Covalent interactions of ATO with the coordinating cysteines result in a pronounced reorganization of the ABP. Most mutants were stabilized by ATO in this study, but only some were transcriptionally rescued. In addition, ATO was effective in one patient-derived xenograft of non-small cell lung carcinoma expressing the R282W mutant, as well as in two cell line-derived xenografts expressing the R175H mutant including CEM-C1, a human T acute lymphoblastic leukemia cell line. 

Niclosamide: Niclosamide is an oral salicylanilide derivative approved by the FDA since 1960 as a drug that acts against infections caused by parasitic worms or helminths (Anthelmintic drug) [121,122]. Characterized by a mild mitochondrial uncoupling activity, this drug has an excellent safety profile in humans as transient and mild mitochondrial uncoupling is tolerable in normal cells [123,124]. Several reports suggest that niclosamide has anticancer activities by impeding various growth promoting pathways such as mTOR, STAT3 and NF-kB [125,126,127,128,129,130]. More recently, niclosamide showed a noteworthy anticancer activity in p53-deficient and -mutant ovarian cancer cells. Surprisingly, wild type p53 cells were less vulnerable to niclosamide-induced mitochondrial uncoupling [131]. These findings, if confirmed in other mutant p53 models, could lead to the repositioning of niclosamide as an anticancer drug that specifically targets TP53-mutated tumors. 

Synthetic lethality strategies: In the area of cancer biology, an oncogenic mutation or a tumor suppressor defect may offer cancer cells a survival advantage based on a secondary survival signal [132]. Accordingly, cancer cells become vulnerable if this secondary survival signal is targeted. Accumulating evidence revealed that TP53 gene mutations provide an opportunity for achieving synthetic lethality in cancer cells. Several studies support this notion by showing that inactivation of the G2 or S checkpoint-associated pathways (i.e., ATR/CHK1, ATM/CHK2, p38MAPK/MK2, WEE1) is synthetic lethal to p53-deficient cancer cells [133,134]. For instance, ATR inhibition induced synthetic lethality in TP53-deficient chronic lymphocytic leukemia cells [135]. Similarly, treatment with the ATR inhibitor AZ-20 increased sensitivity to etoposide in p53-null and R273 mutant cells, but had a lesser impact on p53 wild type cells [136]. Furthermore, recent phase 1 and 2 clinical trials demonstrated that AZD1775, a potent and selective WEE1 inhibitor, enhanced tumor response to carboplatin in TP53-mutated ovarian cancer patients who were refractory to first-line treatment [137,138]. Therefore, the vulnerability caused by TP53 mutations in cancer cells can be efficiently targeted as a strategy to treat patients.

Targeting mutant p53-derived neoantigens: p53 mutants are degraded by the proteasome: the resulting peptides are processed and presented by human leukocyte antigen (HLA) molecules on the cell surface as neoantigens that are recognized by T cell receptors (TCRs) [139,140]. HMTEVVRHC peptide (mutant amino acid underlined), derived from p53-R175H mutant, binds to the HLA allele A*02:01 that is present in more than 40% of U.S. Caucasians [139,141]. Very recently, Han-Chung Hsiue et al. described the development of a TCR-mimic antibody highly specific to the HLA A*02:01-restricted p53-R175H-derived neoantigen. A bispecific version can lyse R175H-mutated cancer cells in a fashion that is dependent on the presence of the neoantigen, both in vitro and in vivo [142]. This novel protein-based approach is very promising and could rapidly enter clinical development.

**Table 1 biomedicines-11-01152-t001:** List of drugs targeting p53 mutants used in clinical trials.

Drug	Mechanism of Action	Trial	Cancer Type	Results	Reference
APR-246	Reconforming agent	NCT04383938	Advanced solid tumor (bladder, gastric, NSCLC, urothelial)	Phase Ib Combination with Pembrolizumab	Park H et al. [143]
		NCT03072043	MDS/AML	Phase II Plus Azacitidine Response rates: MDS 73% Oligoblastic AML 64%	Sallman D et al. [109]
		NCT03588078	MDS/AML	Phase II Plus Azacitidine Response rates: MDS 62% AML 33%	Cluzeau T et al. [110]
		NCT03931291	MDS/AML	Phase II Plus Azacitidine Maintenance therapy following transplant 1-year survival 78.8%	Mishra A et al. [144]
		NCT03745716	MDS/AML	Phase III Plus Azacitidine vs. Azacitidine No significant difference for primary endpoint	Press release
		NCT04214860	MDS/AML	Phase I Plus Azacitidine & Venetoclax	Trial completed Not reported
Arsenic trioxide	Reconforming agent	NCT03855371	MDS/AML	Phase I Plus Decitabine	Currently enrolling
Ganetespib (STA-9090)	HSP90 inhibitor Mutant p53 degradation	NCT02012192	High-grade platinum-resistant ovarian cancer	Phase I/II Combined with Paclitaxel Safe use of the combination	Ray-Coquard et al. [145]
Atorvastatin	Mutant p53 degradation	NCT03560882	MDS/AML Solid tumors	Pilot trial	Currently enrolling
Vorinostat (SAHA)	HDAC inhibitor Mutant p53 degradation	NCT02042989	Metastatic solid tumors	Phase I Plus Ixazomib (proteasome inhibitor) Median survival 7.3 months	Wang Y et al. [146]
		NCT01339871	Metastatic solid tumors	Phase I Plus Pazopanib (VEGF inhibitor) Median survival 12.7 months	Wang Y et al. [146]
Adavosertib (AZD1775/ MK-1775)	Wee1 inhibitor Synthetic lethality	NCT01164995	Refractory ovarian cancer	Phase II Plus Carboplatin Median survival 12.6 months	Leijen S et al. [138]
		NCT03668340	Recurrent uterineserous carcinoma	Phase II Monotherapy Median PFS 6.1 months Median response 9.0 months	Liu et al. [147]

## 6. Conclusions and Future Perspectives

As previously discussed, the most important lesson we learnt from repeated sequencing studies of human samples is that TP53 mutations are generally very early events in the leukemia development process (although this remains uncertain in lower risk MDS with isolated del 5q) (Figure 1 and Figure 2). From a biochemical perspective, two mechanisms can possibly explain the acquisition of a TP53 mutation in a somatic cell. (1) The induction of p53-mediated apoptosis in response to various stresses may lead to the selection of precancerous cells that harbor a TP53 mutation. In addition, cell competition between hematopoietic stem cells (HSCs), a phenomenon controlled by p53 [148,149], could also favor mutated cells [62]. (2) Alternatively, the TP53 gene locus could be associated with DNA fragile sites that are highly sensitive to genomic instability [150,151,152]. Genomic instability can be detected in precancerous lesions prior to the detection of mutations in tumor suppressor genes such as TP53 [153], and is likely to arise from carcinogen- or oncogene-induced replication stress during the formation of solid tumors. This replication stress triggers p53-dependent cell death at the early stage of cancer, but leads to the development of TP53 mutations that further accelerates genome instability and chromothripsis as the cancer progresses to a later stage [153]. For instance, the hotspot p53 mutants R248W and R273H were found to interact with the MRN (Mre11-Rad50-NBS1) complex and to prevent its association with the DNA double-strand breaks, consequently leading to amplified replication stress and impaired DNA damage response [154]. In this second scenario, p53 mutants would act as “supporters” rather than “sources” of genome instability in sporadic cancers. 

In AML, mutant p53 probably initiates leukemic transformation in HSCs (i.e., acting as the source) and the dominant-negative effect may be important at this very early stage. This step is followed by chromosomal losses and “catastrophic” rearrangements [49]. When patients are diagnosed with overt disease, the mutant protein is highly expressed in leukemic cells whereas the wild type allele is always lost among many other tumor suppressors. However, it is still not clear today whether any mutant gain-of-function activities are indispensable for leukemia maintenance at this late stage. 

Therefore, in order to gain better understanding on the roles a missense mutant p53 plays in leukemogenesis, one should analyze in detail the following steps: (1) the acquisition of the mutation in HSCs; (2) the interaction of mutant p53 with wild type p53 when both proteins are still present in the same HSC; (3) the genome instability steps that lead or accompany the leukemia process. For instance, in the case of nonsense mutants, it is unknown whether the sole deletion of the second allele is sufficient to drive leukemogenesis as there might be a need to delete other genes on 17p [97]; and finally (4) the pathways that maintain leukemia. From a clinical point of view, we prioritize the investigation of the last step as patients are seen at the stage of overt disease. Engineered cell lines may not offer the appropriate tools to investigate mutant p53 functions as this approach disregards the mutant p53-dependent multistep process that leads to leukemia. Animal model should be preferable in this setting as compound transgenic genotypes combined with conditional gene expression in the hematopoietic system can help dissect this multistep process. 

Overall, the main difficulty to study mutant p53 functions in acute leukemias is inherent to the leukemia development process that is deemed to be stochastic in this specific setting. Different AML patients with the same TP53 missense mutation may not have biologically equivalent diseases. In addition, different mutants might have different oncogenic functions–if any–at a late stage. Our focus for drug discoveries should probably be on targeting mutant p53 degradation/depletion and on generic cellular processes that are important for the maintenance of aneuploidy in leukemic cells such as DNA replication and chromosome segregation.

## Figures and Tables

**Figure 1 biomedicines-11-01152-f001:**
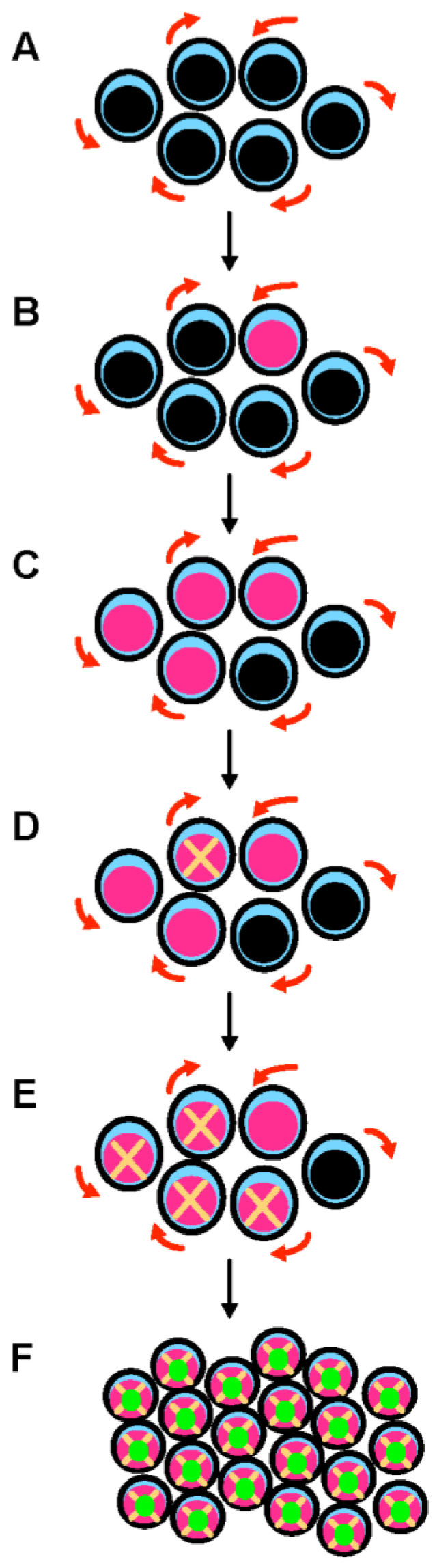
A multistep process leads to the development of TP53-mutated myelodysplastic syndrome and acute myeloid leukemia. (**A**) Hematopoietic stem cells (HSCs) are the only cells within the hematopoietic system that possess the potential for both multipotency (i.e., the potential to differentiate into all of the mature blood cell type) and self-renewal (i.e., the potential to make more stem cells, thus perpetuating the stem cell pool throughout life). Self-renewal is illustrated with red arrows. (**B**) A mutation in the TP53 gene can emerge in a hematopoietic stem cell (pink nucleus). This mutation can appear spontaneously and is selected in response to various stresses, including exposure to cytotoxic treatments. This mutation can be induced by these cytotoxic treatments (such as chemotherapy or radiation therapy) or by workplace exposure to toxic chemicals and carcinogenic substances such as benzene. (**C**) Mutant p53 confers a competitive advantage in the stem cell compartment. At this early stage, cells express both wild type and mutant p53 proteins (potential dominant-negative effect). (**D**) TP53-mutated HSCs acquire chromosomal changes and gene mutations that either result from mutant p53-related genomic instability and chromothripsis, or are induced by cytotoxic chemotherapy/radiation. (**E**) These acquired genetic changes further enhance the fitness of TP53-mutated HSCs. (**F**) Further chromosomal changes lead to the loss of the wild type TP53 allele by 17p deletion or monosomy 17. Loss of wild type TP53 functions and potential gain-of-function activities of mutant p53 are responsible for the development of overt acute myeloid leukemia.

**Figure 2 biomedicines-11-01152-f002:**
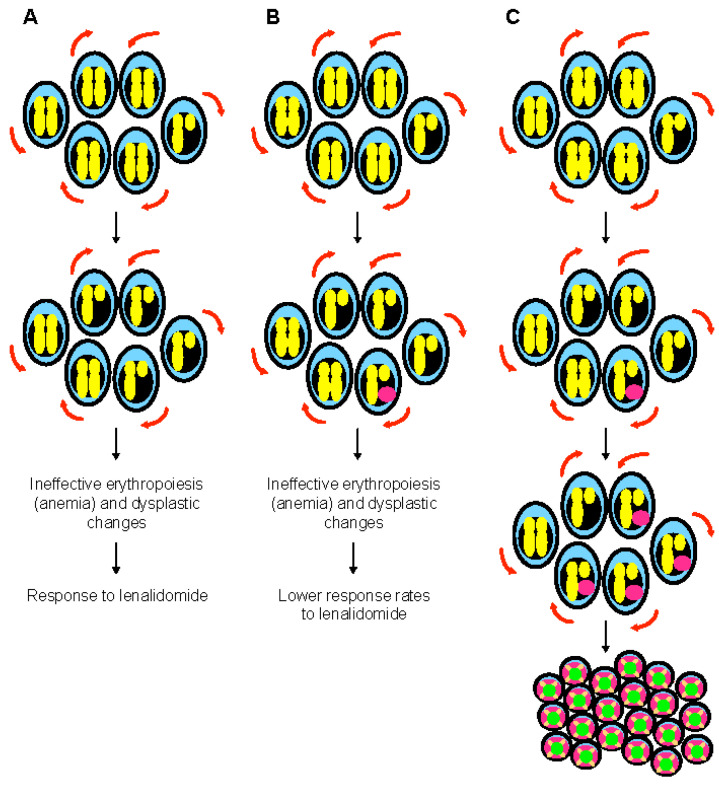
The special case of myelodysplastic syndrome with 5q deletion. (**A**) Deletion of 5q chromosome is somatically acquired and heterozygous. This chromosomal abnormality is present in the hematopoietic stem cell compartment and can be found in all lineages. The clinical phenotype of 5q- syndrome (i.e., ineffective erythropoiesis and dysplastic changes) is related to haploinsufficient gene expression of several genes such as RPS14, APC, and EGR1 [67]. Self-renewal is illustrated with red arrows. (**B**) Monoallelic TP53 mutations are seen at diagnosis in almost 20% of patients with 5q- syndrome, generally at low variant allelic frequency. These mutations are associated with resistance to lenalidomide, but have generally a limited impact on survival. At this stage, karyotype is non-complex [43,44,45,46]. (**C**) Monoallelic TP53 mutations are associated with genomic instability and chromothripsis in myelodysplastic syndrome with isolated 5q deletion. Progression to higher risk myelodysplastic syndrome and acute myeloid leukemia is preceded by acquisition of a complex karyotype, including 17p deletion and biallelic TP53 inactivation [65,66].

## Data Availability

Not applicable.

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
