# Peer review of "TP53 Alterations in Myelodysplastic Syndromes and Acute Myeloid Leukemia"

_biomedicines, 2023, doi:10.3390/biomedicines11041152_

Round 1

Reviewer 1 Report

This is a very interesting review about the occurrence of p53 mutations in myelodysplastic syndrome and acute myeloid leukemia for diagnostic and therapeutic purpose.

The review is well written and organized.

-The review could be improved a little be by including in the introduction section some explanation of the origin of the myelodysplastic syndrome and acute myeloid leukemia.

-The authors should also include a Conclusion and/or future perspective section.

Author Response

Dear reviewer,

We would like to thank you for your review. Your suggestions were taken into consideration in the revised version of the manuscript. We added an introduction and a conclusion/future perspectives sections. We also added two illustrative figures.

Best.

Ramy Rahmé & Pierre Fenaux

Reviewer 2 Report

The review written by Ramy Rahmé et al., entitled “TP53 Alterations in Myelodysplastic Syndromes and Acute Myeloid Leukemia” is an overview on the role of TP53 alterations in MDS/ALM and potential drugs which are currently under investigation in clinical studies to restore the function of mutant p53 in these malignancies.

Whereas many scientific works have demonstrated mutant p53 activities in solid tumors, only few studies have addressed these functions in AML.

In general, the manuscript is informative and gives a detailed overview of p53 modifications and activities in MDS/ALM in sense of mutant p53 and other linked modifications. However, there are other alterations of p53, and I would suggest to at least mention p53 isoforms which are often differentially expressed in human tumors.  

In the text the authors mention several drugs/treatments which are not very common and it would be useful to explain the type and target of particular drug (e.g. Venetoclax, lenalidomide).

I do not see the connection of Magrolimab with p53.

Also, I suggest to describe in few words MCL1, chromothripsis, Anthelmintic drugs,

Lane 84: please add small TP53 “mutant” clones

Lane 249: please add R273 “mutant” cells.

By adding a scheme figure, the manuscript would benefit.

Author Response

Dear Reviewer,

We would like to thank you for your valuable review. We made the following modifications according to your suggestions:

# Lanes 84 and 249: corrected as requested.

# In the phase 1/2 trial that was cited in the review, the triple combination of azacitidine, venetoclax and magrolimab gave very promising results in TP53-mutated AMLs. Overall response and remission rates were added in the manuscript accordingly - 74% and 41%, respectively. Those rates are very encouraging for the development of magrolimab in these cases, and were added in the review.

# Explanations were added and referenced when needed for "venetoclax", "lenalidomide", "MCL1", "Chromothripsis" and "Antihelminthic drug".

# The number of words was increased by adding an introduction and a conclusion/future perspectives sections. We also added two illustrative figures.

Respectfully.

Ramy Rahmé and Pierre Fenaux

Reviewer 3 Report

The authors of this review have discussed the important roles of the TP53 gene in myelodysplastic syndromes and acute myeloid leukemia. They have described the characteristics of TP53 mutations, potential cellular pathways affected by these mutations, and current treatment strategies for these conditions.  To improve accessibility for non-expert readers, the authors could provide more general information about TP53 at the beginning of the review and summarize the key points at the end. Overall, this review article is well-written and informative, and with some minor revisions, it would be suitable for publication.

·         At the beginning of the review, please consider including more background information of TP53, for example, its functions and its roles as a tumor suppressor.

·         Please complete author contributions, funding sources and conflicts of interest.

·         It would be beneficial to include a discussion section at the end of the review that summarizes the current advances, challenges, and future directions for studying TP53-related pathways in myelodysplastic syndromes (MDS) and acute myeloid leukemia (AML). This would help provide a comprehensive understanding of the research in this area and highlight the key areas that require further investigation.

Author Response

Dear Reviewer,

We would like to thank you for your valuable review. The manuscript was improved by adding an introduction and a conclusion/future perspectives section. We also added two illustrative figures.

Author contributions, funding sources and conflicts of interest: completed as requested.

We hope the manuscript has been improved accordingly.

Respectfully.

Ramy Rahmé and Pierre Fenaux